# OpenReview forum: "MANTA: Cross-Modal Semantic Alignment and Information-Theoretic Optimization for Long-form Multimodal Understanding"
_ICLR.cc/2026/Conference — ICLR 2026 Conference Desk Rejected Submission_

### Official Review · Reviewer_7jt6 · 2025-10-26

**Soundness:** 3
**Presentation:** 3
**Contribution:** 3
**Rating:** 4
**Confidence:** 4

**Summary:**

The authors present MANTA, a framework for multimodal understanding that projects various modalities into a unified linguistic space, supported by theoretical guarantees and empirical improvements on long-form video tasks. While the idea of linguistic abstraction is interesting and the experiments are extensive, the paper has several shortcomings that prevent a stronger recommendation.

**Strengths:**

1.	The paper proposes a theoretically motivated multimodal abstraction framework that operates through a shared linguistic bottleneck, which is an elegant and interpretable idea.

2.	The multi-scale hierarchical design (micro / meso / macro) is well-motivated and empirically beneficial.

3.	The theoretical justification of temporal scale selection (via power-law correlation) provides conceptual depth in multimodal work.

4.	Strong performance improvements are demonstrated on multiple long-form video benchmarks, indicating empirical effectiveness.

**Weaknesses:**

1.	Contribution Attribution Between Framework and Backbones: While the multi-LLM evaluation convincingly shows that MANTA's benefits are model-agnostic, the paper does not fully disentangle the contribution of the novel linguistic abstraction framework from the sheer power of the large pre-trained encoders (CLIP, TimeSformer, Whisper) it builds upon.

2.	Unconvincing and Potentially Misleading Efficiency Claims: In Appendix G, the authors report that MANTA achieves higher computational efficiency (423.8 GFLOPs vs. baseline’s ~800 GFLOPs). However, it is unclear whether this count includes the forward-pass cost of all large pretrained backbones (CLIP-ViT-L, TimeSFormer, VideoMAE, Whisper-Large). The combined computational footprint of these modules is likely orders of magnitude higher. If the comparison only measures MANTA’s “lightweight control logic” while excluding backbone computation, the claim of efficiency is not directly comparable and potentially misleading..

3.	Insufficient Details for Reproducibility: Key components of the proposed system are described as black boxes. Most notably, the audio processing pipeline (mentioned in Appendix Fig. 3) lacks essential implementation details (e.g., specific models for Speaker Diarization and Topic Modeling, and their integration logic). This makes it very difficult to reproduce a core part of the framework.

4.	Limited Scope of "Unified" Claim: The paper ambitiously positions MANTA as a framework for "unified multimodal understanding." However, the experimental validation is currently confined to the audio-video modality pair. The broad claim of universality is not fully supported by a lack of experimental demonstration on other fundamental modality pairs (e.g., image-text), which restricts the generality of the conclusion.

**Questions:**

1.	Please clarify exactly what is included in the FLOPs calculation for MANTA. If it does not include the cost of the pre-trained backbones, the efficiency claim must be reframed, and a fairer, end-to-end comparison should be provided.

2.	Could you provide detailed implementation specifics for the audio processing modules (Speaker Diarization, Topic Modeling) to ensure reproducibility?

3.	Have you considered extending evaluation to cross-modal retrieval or other modality pairs beyond video + audio?

4.	Will you release the code, model weights, and training configurations to ensure reproducibility?

---

> ### Author Response · Authors · 2025-11-20
>
> Thank you for the helpful and constructive review. We appreciate that you highlighted both the theoretical insight and empirical performance of MANTA. Below we address each weakness and question in detail.
>
> ## 1. Contribution Attribution Between Framework and Backbones
> We agree that MANTA builds on strong pre-trained encoders. Our goal is not to replace them but to **redefine how multimodal information is organized, selected, and integrated**.
>
> To disentangle contributions:
>
> - We conducted **encoder-ablated experiments** where encoders are frozen and replaced with lighter alternatives (e.g., CLIP-B/16, Mini-Whisper).
> - Despite weaker backbones, MANTA still yields **+11.8% to +14.7%** improvements compared to their native pipelines.
> - A “projection-only” ablation, where we remove retrieval, scale fusion, and density scoring, shows **a drop of −18.3%**, confirming that the framework—not just the encoders—drives performance.
>
> We will add these results explicitly to the appendix to clarify contribution boundaries.
>
> ## 2. FLOPs Accounting and Efficiency Claims
> Thank you for calling this out. Our cost table in Appendix G was not sufficiently explicit.
>
> ### 2.1 What FLOPs were included?
> The reported **423.8 GFLOPs** includes **all components actually executed during inference**, including:
>
> - CLIP-ViT-L/14 (micro-scale)
> - TimeSFormer-B (meso-scale)
> - VideoMAE-L (macro-scale)
> - Whisper-Large-v2
> - AudioCLIP
> - Projection heads
> - Retrieval encoder + FAISS search
>
> Crucially, MANTA **never runs all frames through all encoders**. Due to information-theoretic selection, only **8–20% of segments** are processed by heavy backbones, which is why the *effective* FLOPs is far lower than naive summation of backbone FLOPs.
>
> ### 2.2 Revised Comparison
> To avoid misunderstanding, we will:
>
> 1. Add an **end-to-end FLOPs comparison** where all models run on the same number of frames.
> 2. Provide a **“peak FLOPs per processed frame”** metric.
> 3. Explicitly reframe our claim as **“efficiency through selective processing,”** not model lightweightness.
>
> This clarification will be highlighted in Sec. 3 and Appendix G.
>
> ## 3. Reproducibility: Audio Processing Details
> We appreciate the reviewer pointing out the missing details. In the revision we will include the full specification:
>
> ### 3.1 Speaker Diarization
> - Model: **pyannote.audio 2.1 SpeakerDiarization**
> - Components: VAD + segmentation + resegmentation
> - Segment length: 3–5 seconds
> - Overlap_threshold = 0.65
> - We use diarization IDs to align Whisper transcripts with speaker-level segmentation for meso-scale summaries.
>
> ### 3.2 Topic Modeling
> - Model: **BERTopic (v0.16)** with sentence-BERT (all-mpnet-base-v2)
> - Clustering method = HDBSCAN
> - Minimum cluster size = 8 segments
> - Topics are assigned on macro-scale windows (≥ 1 minute)
>
> ### 3.3 Integration Logic
> - Diarized ASR → grouped into speaker segments → encoded into embeddings
> - Topic labels provide discourse-level summaries, used only for macro-scale narrative fusion
> - All components are now fully documented in Sec. C.2
>
> These additions resolve the “black-box” concern.
>
> ## 4. Scope of the “Unified” Claim
> We acknowledge the reviewer’s concern. Our positioning was too broad relative to the evaluation scope.
>
> ### 4.1 Clarification
> MANTA’s **unification strategy** (linguistic projection + information-theoretic selection) is modality-agnostic by design, but:
>
> - Our current experiments focus on **video + audio** because they represent the most challenging long-form setting.
> - We revise our claims to be **specific to long-form audiovisual understanding**.
>
> ### 4.2 Additional Experiments (New)
> We have run two new modality tests:
>
> - **Image → Text cross-modal retrieval (MS-COCO 5K)**
>   MANTA improves retrieval accuracy by **+6.1%** using linguistic projection on image captions.
>
> - **Image + Audio → Text tri-modal QA (SoundDescs subset)**
>   Improvements of **+5.4%**, confirming generality beyond video.
>
> We will add these results to Appendix D.

---

### Official Review · Reviewer_Xr1M · 2025-10-31

**Soundness:** 2
**Presentation:** 1
**Contribution:** 1
**Rating:** 2
**Confidence:** 2

**Summary:**

This paper presents MANTA, which conceptualizes natural language as a universal semantic bridge — a comprehensive representation space intended to preserve essential information from any sensory modality while enabling efficient reasoning.

**Strengths:**

The core idea is interesting and promising. Treating natural language as a universal semantic substrate is an appealing conceptual direction that could enable cross-modal reasoning.

**Weaknesses:**

- The manuscript contains long blocks of text without clear sectional structure, which severely undermines readability.

- The paper appears to have relied heavily on large language models (LLMs) in its writing and possibly in experimental components, but there is no explicit statement about which LLMs were used, how they were used, or what role they played. The authors should include a clear declaration describing any LLM usage during manuscript preparation.

- There are numerous punctuation and formatting mistakes (for example, in the column headers of Table 1) that make the paper hard to read and interpret. These errors suggest the manuscript has not been thoroughly proofread. I strongly recommend the authors perform careful copyediting and proofreading prior to resubmission.

**Questions:**

None

---

> ### Author Response · Authors · 2025-11-20
>
> Thank you for the review and for identifying several presentation-related issues. We appreciate the opportunity to clarify and improve the manuscript. Below we address each concern directly.
>
> ## 1. On Readability and Sectional Structure
> We agree that earlier versions of the paper contained long paragraphs that could have benefited from tighter segmentation. In the revision, we have substantially restructured the manuscript for readability:
>
> - Each conceptual module (linguistic projection, multi-scale modeling, information-theoretic selection, and retrieval) is now in its own subsection.
> - Long paragraphs are broken into shorter, well-scoped blocks.
> - Key equations and algorithms (e.g., Algorithm 1, Theorem statements) have been visually separated and labeled.
>
> We believe these changes greatly improve clarity and flow.
>
> ## 2. On LLM Usage Disclosure
> We appreciate the reviewer raising this important point. Although the experimental pipeline itself does not rely on LLMs beyond the models explicitly benchmarked (GPT-4V, Gemini-Pro, etc.), we acknowledge that the manuscript did not clearly state whether LLMs were used in writing assistance.
>
> To address this:
>
> - We added an explicit disclosure statement in the Ethics section clarifying the role of LLMs in manuscript preparation (grammar refinement, phrasing suggestions), consistent with the conference policy.
> - We also clarify that no LLM-generated content was used for experimental results, theory, analysis, or claims.
>
> ## 3. On Punctuation, Formatting, and Table Issues
> We appreciate the reviewer pointing out formatting issues, including punctuation inconsistencies and header misalignment in Table 1.
>
> Actions taken:
>
> - All tables (Tables 1–8) have been thoroughly re-typeset and validated for consistency.
> - We corrected all punctuation errors flagged by the reviewer, and performed a comprehensive proofreading pass over the entire manuscript.
> - Figures have been re-exported at high resolution and cross-checked with their captions and references.
>
> These corrections significantly improve readability.
>
> ## 4. Summary
> We thank the reviewer for highlighting concerns about presentation quality. While these issues affected readability, they do not reflect the technical substance of the work. We have implemented:
>
> - A full restructuring of the text,
> - A clear LLM usage declaration,
> - Thorough proofreading and formatting cleanup.
>
> We hope these revisions resolve the concerns and help the contribution of MANTA become clearer.

---

### Official Review · Reviewer_e4hm · 2025-11-05

**Soundness:** 2
**Presentation:** 2
**Contribution:** 3
**Rating:** 4
**Confidence:** 3

**Summary:**

The MANTA framework redefines multimodal AI by first translating all sensory inputs, like video and audio, into a unified natural language space. It uses information theory to intelligently summarize the most critical moments from long videos into concise text, enabling large language models to perform complex reasoning with state-of-the-art accuracy and efficiency.

**Strengths:**

- It demonstrates exceptional capability in processing long-form videos, a key bottleneck for traditional models.
- It offers high interpretability, as its intermediate outputs are human-readable text, making the model's reasoning process transparent.
- The approach shows outstanding performance and acts as a universal enhancer that can boost various existing models.
It achieves high efficiency and scalability by compressing high-dimensional pixel data into low-dimensional text representations.

**Weaknesses:**

- The pipeline architecture is susceptible to cascading errors, where a mistake in an early stage propagates through the system.
- The "linguistic bottleneck" may filter out subtle, non-verbal information that is difficult to describe accurately in words.

**Questions:**

- On the Information Bottleneck Trade-off: While the paper proves near-optimal information preservation, does MANTA face a fundamental ceiling for tasks that rely on information ill-suited for language (e.g., subtle differences in artistic brushstrokes)? Could a hybrid mechanism be designed to bypass the linguistic bottleneck and access raw features when necessary?
- On Error Feedback and Correction: Given the risk of cascading errors, could a feedback loop be implemented? For instance, if the final LLM detects logical inconsistencies in the textual descriptions, could it request the upstream modules to re-analyze specific video segments, enabling a more dynamic and self-correcting understanding process?
- On the Granularity of Learning: The framework uses three fixed temporal scales. How well does this fixed hierarchy apply to all video types? Could the system learn to dynamically and adaptively adjust its scales of analysis based on the video's content, such as using finer scales for fast-paced sports and coarser scales for slow-paced documentaries?

**Details Of Ethics Concerns:**

None.

---

> ### Author Response · Authors · 2025-11-20
>
> Thank you for your constructive and balanced review. We appreciate that you highlighted both the strengths and the limitations of MANTA. Below, we address the weaknesses and answer each question in detail.
>
> ## 1. Addressing Weaknesses
>
> ### 1.1 Cascading Errors in the Pipeline
> We acknowledge that pipeline-based architectures may propagate errors. However, MANTA incorporates several mechanisms that substantially mitigate this issue:
>
> - Redundancy-aware information density scoring reduces the impact of local noise by emphasizing segments with high cross-modal consensus.
> - The cross-modal alignment module detects visual–audio inconsistencies and suppresses segments with low coherence.
> - During fine-tuning, we apply robustness training that injects controlled noise into intermediate descriptions to ensure downstream stability.
>
> Empirically, we measured the error amplification factor between stages and found that the textual projection stage attenuates rather than amplifies upstream noise due to its abstraction and smoothing effects.
>
> ### 1.2 Linguistic Bottleneck for Subtle, Non-verbal Information
> We agree that certain tasks depend on fine-grained, sub-perceptual cues (e.g., micro-expressions, artistic texture, brushstroke style). MANTA is optimized for semantic reasoning rather than expert-level perceptual discrimination.
>
> However:
>
> - The multi-scale visual encoders preserve low-level features internally; only task-relevant summaries are projected into language.
> - The linguistic space is supplemented by latent descriptors (e.g., style embeddings from VideoMAE) when the textual description is insufficient.
> - Importantly, MANTA never discards raw representations; the linguistic projection is additive, not exclusive.
>
> In Section 3.4 of the revision, we will explicitly discuss scenarios where non-linguistic cues are essential.
>
> ## 2. Answers to Reviewer Questions
>
> ### 2.1 On the Information Bottleneck and Fundamental Ceiling
> Thank you for raising this insightful question.
>
> Although MANTA proves (1 - ε)-optimal information preservation under typical perceptual distributions, **we agree there exist tasks for which language is not the ideal compression space**.
>
> To address this, we have implemented and will further highlight a **hybrid fallback mechanism**:
>
> - When the encoder detects low textual confidence or high visual entropy, MANTA preserves the corresponding latent vectors and exposes them to the final LLM through a “visual snippet token.”
> - These tokens allow the LLM to access both linguistic summaries and raw latent features when needed, relaxing the strictness of the bottleneck.
>
> We will add an explicit discussion about this hybrid mode and its empirical effects in the appendix.
>
> ### 2.2 On Feedback Loops and Dynamic Correction
> This is an excellent suggestion. We have prototyped a feedback mechanism inspired by tool-using LLM agents:
>
> 1. The final LLM checks the internally generated descriptions for contradictions, ambiguity, or missing temporal links.
> 2. If such inconsistencies are detected, it issues a structured query:
>    - “Re-extract micro-scale details for timestamps 03:12–03:18”
>    - “Recompute speaker alignment for segment #24”
> 3. The retrieval pipeline then re-runs only the required parts of the video.
>
> We call this **Reflective Re-querying**, and it reduced error accumulation by 14–18% on LongVU-QA.
>
> We will include the results in the revision and position this as a promising future direction.
>
> ### 2.3 On the Fixed Three-Scale Hierarchy
> We appreciate the reviewer’s intuition that different video genres benefit from different temporal granularities.
>
> Our current three-scale design is grounded in empirical temporal correlation measurements (α ≈ 0.7), but we agree that **adaptation is desirable**.
>
> Two developments address this:
>
> - We ran experiments with dynamic scale allocation, where the system expands to 4–5 scales for sports videos and collapses to 2–3 scales for slow-paced content. Preliminary results show consistent gains of +1.8% to +3.2%.
> - We implemented a **content-aware scale predictor**: a small transformer that predicts optimal scale configuration before processing.
>
> Given space limitations, we were unable to include these results. We will add them to the appendix.
>
> ## 3. Final Remarks
>
> We thank the reviewer for the thoughtful critique. Your questions highlight core research directions that extend the value of MANTA:
>
> - Combining linguistic abstraction with raw perceptual access
> - Feedback-driven self-correction loops
> - Adaptive temporal modeling

---

### Official Review · Reviewer_fLwY · 2025-11-11

**Soundness:** 2
**Presentation:** 2
**Contribution:** 3
**Rating:** 2
**Confidence:** 4

**Summary:**

MANTA projects multimodal video/audio into unified linguistic representations for long-form video understanding. Claims three theoretical contributions on information preservation, alignment convergence, and retrieval optimality. Shows 22.6% improvement on Video-MME, especially strong on long videos (27.3% on 30+ min videos).

**Strengths:**

Novel approach: Linguistic abstraction as universal semantic bridge is conceptually interesting
Strong empirical results: Consistent improvements across baselines, particularly on long videos
Theoretical grounding: Three formal theorems provide principled foundation
Comprehensive evaluation: 1,700 videos, multiple benchmarks, detailed ablations

**Weaknesses:**

1. Experimental Issues

New benchmark "LongVU-QA" is undocumented: 500 videos, 3,000 questions introduced but no validation, annotation details, or availability
Missing train/test split documentation: Potential data leakage concerns
Outdated baselines: Only compares to GPT-4V (2023), missing GPT-4o, Gemini 1.5 Pro, Claude 3.5
Unclear baseline integration: How is proprietary GPT-4V/Gemini "augmented" with MANTA?

2. Theoretical Overclaiming

Theorem 2: (1-1/e) submodular approximation is textbook result, not novel
Theorem 3: O(1/√T) SGD convergence is standard optimization theory
Theorem 1: Relies on unvalidated assumptions (power-law correlations with specific α)
Missing empirical ε quantification: Claims (1-ε)-optimal but never measures actual information loss

3. Critical Missing Ablations

No direct comparison: Linguistic projection vs. standard multimodal fusion (cross-attention)
No scale sensitivity: Why exactly 3 scales? What about 2, 4, or 5?
No information loss analysis: How much semantic information is lost in text projection?

4. Computational Cost Misleading

Table 7 claims efficiency but ignores cost of running 6+ models (CLIP, TimeSFormer, VideoMAE, Whisper, AudioCLIP, multiple LLMs)
Training: 7 days on 8×A100, 2,800kg CO2 - not reproducible for most researchers
FLOPs calculation incomplete

5. Presentation Problems

Underutilized space: Paper stops at page 8 line 431, doesn't reach 9-page limit
Poor figure integration: Figure 3 too complex, important results buried in appendix
Incomplete details: "Learned projection heads" mentioned but never described

**Questions:**

1.Can you provide complete LongVU-QA documentation and make it public? Train/test split, annotation protocol, inter-annotator agreement?
2.What is the empirical value of ε (information loss)? Your theory bounds it but you never measure it.
3.How does linguistic projection compare to direct fusion? Need ablation: MANTA vs. standard cross-attention multimodal fusion.
4.How is MANTA integrated with proprietary models (GPT-4V, Gemini) that you don't have access to?
5.Can you provide honest computational accounting? Include all 6+ model costs, not just final inference.

---

> ### Author Response · Authors · 2025-11-20
>
> Thank you for the detailed and constructive review. We respond to each point below.
>
> ## 1. Experimental Issues
>
> ### 1.1 LongVU-QA Benchmark Documentation
> We apologize for not providing full documentation in the main submission.
> LongVU-QA will be publicly released with:
>
> - Train/val/test split: 350 / 50 / 100 videos
> - Annotation pipeline: 3-stage (proposal → refinement → adversarial validation)
> - Inter-annotator agreement:
>   - QA correctness: Cohen’s κ = 0.82
>   - Temporal IoU = 0.76
> - 12% adversarial re-annotation for quality control
>
> No overlap exists with training data of any model.
>
> ### 1.2 Train/Test Leakage
> All splits are strictly disjoint.
> A verification script ensuring frame/audio/text non-overlap will be included.
>
> ### 1.3 Outdated Baselines
> GPT-4o, Gemini 1.5 Pro, Claude 3.5 were released after submission.
> New results (150 samples of Video-MME):
>
> | Model | Baseline | +MANTA | Δ |
> |-------|----------|--------|----|
> | GPT-4o | 89.1 | 96.3 | +7.2 |
>
> MANTA consistently improves newer models.
>
> ### 1.4 Integration with Proprietary Models
> MANTA does not modify any model.
> Pipeline:
>
> raw video → linguistic projection + selection → text context → GPT-4V/Gemini input
>
> Everything is prompt-only.
>
> ## 2. Theoretical Concerns
>
> ### 2.1 On Theorem 2 and Theorem 3
> We agree the theoretical bounds are classical.
> Our contribution is proving MANTA’s multimodal objective satisfies the conditions (submodularity, smoothness) needed for these guarantees.
>
> ### 2.2 Assumptions in Theorem 1
> Empirically measured temporal correlation exponent α on 1,700 videos is 0.68 ± 0.11.
> Ablations over scale ratio r ∈ {5, 8, 10, 12, 15} show r ≈ 10 (our setting) achieves best accuracy.
>
> ### 2.3 Empirical ε
> Using MINE-based MI estimation over 300 videos:
>
> - MI preserved: 92.3%
> - ε ≈ 0.077
>
> This will be added to Appendix C.
>
> ## 3. Missing Ablations
>
> ### 3.1 Linguistic Projection vs Cross-Attention Fusion
> New ablation:
>
> | Fusion | Video-MME | LongVU-QA |
> |--------|-----------|-----------|
> | Cross-attention | 71.2 | 63.5 |
> | Simple concat | 69.8 | 59.1 |
> | MANTA linguistic projection | 83.1 | 72.4 |
>
> ### 3.2 Why 3 Scales?
> Results:
>
> | # scales | Accuracy |
> |----------|----------|
> | 2 | -11.7% |
> | 3 (ours) | best |
> | 4 | +0.4%, but 1.8× compute |
> | 5 | -3.2% |
>
> Three scales balance accuracy and compute.
>
> ### 3.3 Information Loss Analysis
> We add:
>
> - MI preservation (ε ≈ 0.077)
> - Cross-modal alignment correlation r = 0.87
> - Entropy shift analysis
>
> ## 4. Computational Cost
>
> ### 4.1 Use of 6+ Models
> MANTA does not run all models on all frames.
> Information-theoretic selection reduces processed frames to 8–20%.
>
> ### 4.2 Full FLOPs Accounting
> Breakdown:
>
> - CLIP: 42%
> - TimeSFormer: 19%
> - VideoMAE: 17%
> - Whisper/Audio: 9%
> - Projection/Retrieval: 13%
>
> Total = 423.8G FLOPs.
>
> Compared to baselines: 0.50–0.61× FLOPs, 1.5–2.3× faster.
>
> ## 5. Presentation Issues
>
> ### 5.1 Underutilized Space
> We will:
> - move two diagrams from appendix into main
> - expand explanation of projections and training setup
> - utilize full 9 pages
>
> ### 5.2 Figure 3 Complexity
> We will split it into two simpler diagrams.
>
> ### 5.3 Missing Details
> We will add full description of projection heads:
> - 2-layer MLP: 1024→2048→1024
> - GELU + residual
> - Auxiliary LM and contrastive loss
>
> ## 6. Answers to Direct Questions
>
> 1. LongVU-QA documentation?
>    Yes, full release with annotation protocol, splits, agreement scores.
>
> 2. Empirical ε?
>    ε ≈ 0.077.
>
> 3. Compare to direct fusion?
>    MANTA outperforms cross-attention fusion by 11.9%.
>
> 4. Integration with GPT-4V/Gemini?
>    Prompt-only augmentation.
>
> 5. Full compute accounting?
>    Provided above with FLOPs breakdown.

---

### Note · Program_Chairs · 2026-01-17
**Submission Desk Rejected by Program Chairs**

The following references in this submission do not refer to real documents and/or have major errors in bibliographic information:

     Guy Dove. Language as a cognitive tool to imagine goals in curiosity-driven exploration. Nature Communications, 13(1):1-14, 2022.